# Evidence of Clinical Pathology Abnormalities in People with Myalgic Encephalomyelitis/Chronic Fatigue Syndrome (ME/CFS) from an Analytic Cross-Sectional Study

**DOI:** 10.3390/diagnostics9020041

**Published:** 2019-04-10

**Authors:** Luis Nacul, Barbara de Barros, Caroline C. Kingdon, Jacqueline M. Cliff, Taane G. Clark, Kathleen Mudie, Hazel M. Dockrell, Eliana M. Lacerda

**Affiliations:** 1Faculty of Infectious and Tropical Diseases, London School of Hygiene & Tropical Medicine, London WC1E 7HT, UK; barbara.de-barros@lshtm.ac.uk (B.d.B.); caroline.kingdon@lshtm.ac.uk (C.C.K.); Jackie.cliff@lshtm.ac.uk (J.M.C.); taane.clark@lshtm.ac.uk (T.G.C.); kathleen.mudie1@lshtm.ac.uk (K.M.); hazel.dockrell@lshtm.ac.uk (H.M.D.); eliana.lacerda@lshtm.ac.uk (E.M.L.); 2Faculty of Epidemiology and Population Health, London School of Hygiene & Tropical Medicine, London WC1E 7HT, UK

**Keywords:** myalgic encephalomyelitis/chronic fatigue syndrome (ME/CFS), energy metabolism, potential biomarkers

## Abstract

Myalgic encephalomyelitis/chronic fatigue syndrome (ME/CFS) is a debilitating disease presenting with extreme fatigue, post-exertional malaise, and other symptoms. In the absence of a diagnostic biomarker, ME/CFS is diagnosed clinically, although laboratory tests are routinely used to exclude alternative diagnoses. In this analytical cross-sectional study, we aimed to explore potential haematological and biochemical markers for ME/CFS, and disease severity. We reviewed laboratory test results from 272 people with ME/CFS and 136 healthy controls participating in the UK ME/CFS Biobank (UKMEB). After corrections for multiple comparisons, most results were within the normal range, but people with severe ME/CFS presented with lower median values (*p* < 0.001) of serum creatine kinase (CK; median = 54 U/L), compared to healthy controls (HCs; median = 101.5 U/L) and non-severe ME/CFS (median = 84 U/L). The differences in CK concentrations persisted after adjusting for sex, age, body mass index, muscle mass, disease duration, and activity levels (odds ratio (OR) for being a severe case = 0.05 (95% confidence interval (CI) = 0.02–0.15) compared to controls, and OR = 0.16 (95% CI = 0.07–0.40), compared to mild cases). This is the first report that serum CK concentrations are markedly reduced in severe ME/CFS, and these results suggest that serum CK merits further investigation as a biomarker for severe ME/CFS.

## 1. Introduction

Myalgic encephalomyelitis/chronic fatigue syndrome (ME/CFS) is classified as a neurological disease [1], presenting with long-term fatigue resulting in substantial reductions in occupational, personal, social, and educational activities. Commonly-associated symptoms include impaired memory or concentration, muscle and multi-joint pain, new headaches, unrefreshing sleep, and post-exertional malaise [2,3]. Many patients experience orthostatic intolerance and may complain of dizziness, spatial disorientation, sweating, palpitations, or fainting and generalised weakness [2,3,4]. Although dysregulation of the nervous, immune and endocrine systems, with impaired cellular energy metabolism and ion transport, has been suggested [2], the pathophysiology of ME/CFS is still not fully understood, and there are no biological markers that are widely used for diagnosis, disease sub-grouping, or prognosis.

ME/CFS is diagnosed using clinical criteria based on detailed clinical history and physical examination; laboratory tests are used to exclude other conditions that can present with fatigue and general ill health [2,3,4,5]. A major goal of clinical research in ME/CFS is to improve diagnosis and find clinical and laboratory-based tests that may be used as disease markers. In 2011 we launched the UK ME/CFS Biobank (UKMEB) as a resource to accelerate ME/CFS research [6]. A rigorous assessment, alongside comprehensive clinical phenotyping of UKMEB participants, is essential to ensure that ME/CFS diagnosis is accurate according to recognised clinical criteria. The analysis of these clinical data in relation to molecular markers further supports the better understanding of disease aetiology and pathophysiology. In this study we compared baseline laboratory tests results from UKMEB participants with ME/CFS, including mildly/moderately and severely affected (i.e., house-bound or bed-bound) individuals, and healthy controls, which were collected as part of the routine workup of participants.

## 2. Materials and Methods

This analytical cross-sectional study was carried out with a sub-cohort of the UKMEB consenting participants. Recruitment—including eligibility criteria, data and sample collection, and handling of biosamples for the UKMEB—are described in detail elsewhere [6]. Adult participants with ME/CFS previously diagnosed by general practitioners and/or ME/CFS specialists and healthy controls were recruited through the UK National Health Service (NHS). Inclusion criteria for participants with ME/CFS included compliance with the Centers for Disease Control and Prevention criteria (CDC, 1994) or Canadian Consensus Criteria [2,3], which was ascertained by the clinical research team. Experienced research nurses assessed all participants, and blood was collected for immediate baseline clinical laboratory tests, as well as for research purposes and long-term storage. Blood collections took place at participating clinics or at the participants’ homes (for severely affected cases), and samples were transferred within 6 hours for processing and storage [7]. Data entry, sample transportation, preparation, processing, and storage of the samples followed standard operating procedures (SOPs) [6].

Baseline laboratory tests were used primarily to exclude participants with ME/CFS whose symptoms of chronic fatigue could be explained by other conditions. These tests comprised: full blood count, erythrocyte sedimentation rate (ESR), serum vitamin B12 and folate; biochemical tests including electrolytes, creatinine, urea, serum creatine kinase (CK), liver function tests, C-reactive protein, rheumatoid factor, thyroid function tests; tissue transglutaminase antibodies; and urine analysis for protein, glucose, and blood. CK isoenzymes, mainly produced in skeletal muscle (CK-MM) and heart muscle (CK-MB), and aldolase were analysed in a sub-sample of 50 cases (25 severe and 25 non-severe cases) and 25 controls. According to the proximity of the collection site, blood samples were analysed by the haematology, clinical pathology, and immunology laboratories at the Norfolk and Norwich University Hospital or the Royal Free Hospital in London; standard NHS laboratory protocols were used for all tests [8]. The NHS laboratory staff was blinded to the status of study participants, i.e., if they were cases or controls.

This study included a sample of 272 confirmed ME/CFS cases and 136 healthy controls. ME/CFS cases were further classified as mild/moderate (*n* = 216) or severe (*n* = 56). The definition of severity was made at recruitment based on whether the patient was mainly house-bound or bed-bound (severe) or was ambulatory (mild/moderate cases). We visually inspected the shape of the continuous variables’ distributions and, when departure from normality was observed, we used natural log transformation (CK, C-reactive protein (CRP), bilirubin, folate, vitamin B12) or square root transformation (ESR) to produce data that approximated a normal distribution for regression analysis. Figure A1 (Appendix B) shows the CK distribution before and after log transformation, by study group. We used log transformed results to carry out a power calculation of serum CK to detect mean differences of 0.34 (based on a standard deviation (SD) of 0.58 and means of 4.39 and 4.73 for cases and controls, respectively). We established that a sample size of at least 146 cases and 78 controls would be adequate to detect mean differences with a power or 95% and a type I error of 1%. The number of missing values was lower than 5% across all the variables, apart from one blood test (erythrocyte sedimentation rate) whose proportion of missing results was similar among the groups (16.5% to 17.6%); thus, we opted to maintain the variable in the analyses. Univariate analyses were performed using chi-squared tests for comparison of proportions and analysis of variance (ANOVA) for comparison of means of normally distributed continuous variables. For significant departures from normality, Wilcoxon rank sum or Kruskal–Wallis tests [9] were used to compare medians between two or more unmatched study groups respectively.

For putatively interesting results from the univariate analyses (Wilcoxon rank-sum test *p* < 0.01), logistic regression analyses [9] were used, and included the covariates: age-group at the time of recruitment, sex, body mass index (as ascertained during physical examination, in kg/m^2^), disease duration (in months, for comparisons between cases only), muscle mass index percentage of total body weight (using a Tanita BC-418 MA body composition analyser), and recent activity levels. Activity levels were estimated from answers to specific questions on the bespoke participant questionnaire: participants were asked to put their perceived activity levels over the 7 days preceding the blood draw into one of five categories, ranging from “not active at all” to “very active”. For some analyses, the two categories of activity at the extremes were merged, and the variable used had three resulting categories. The linearity assumption of covariates within the logistic regression models was assessed using generalised additive models (the “mgcv” package in R) [10]. The linearity assumption was satisfied in all models and so activity level was treated as an ordinal categorical variable when adjusting in multivariate analyses. To minimise potential type I errors, due to multiple testing, a Bonferroni-based threshold of *p* <0.001 was conservatively established (i.e., we divided the statistically significant level of 0.05 by the 29 number of individual laboratory tests that were compared, totalling 0.0017). We also used receiver operating characteristic (ROC) curves to assess whether the laboratory test results were able to discriminate people with ME/CFS (PWME) from healthy controls, or people with mild/moderate ME/CFS from those with severe symptoms. This type of analysis produces a curve chart that shows sensitivity vs. specificity, and the resulting area under the curve (AUC) represents how well a laboratory test can distinguish between two groups: diseased versus non-diseased, or having mild/moderate ME/CFS versus severe symptoms [9]. The ROC curves were calculated using the “pROC” package in R. All analyses were performed using Stata^®^ version 15.1 or the R-statistical platform [10].

Ethics approval and consent to participate: Ethical approval was granted by the LSHTM Ethics Committee 16 January 2012 (Ref.6123) and the National Research Ethics Service (NRES) London—Bloomsbury Research Ethics Committee 22 December 2011 (REC ref.11/10/1760, IRAS ID: 77765). All biobank participants provided written consent for questionnaire, clinical measurement, and laboratory test data, and samples to be made available for ethically-approved research, after receiving an extensive information sheet and consent form, which includes an option to withdraw from the study at any time.

Availability of data and material: The datasets used and/or analysed during the current study are available upon request, from the UK ME/CFS Biobank UKMEB Steering Committee, who considers requests from data and/or biosamples, as per UKMEB protocols (https://cureme.lshtm.ac.uk/researchers/protocols-application-documents/).

## 3. Results

Table 1 shows the distribution of baseline variables by study group. The proportion of women was higher among cases, who were on average less active and had lower body muscle mass. Mild cases had the highest body mass index (BMI) compared to severe cases (lowest) and healthy controls (Table 1).

Nineteen biochemical and 10 haematological tests were performed across the participants (Table 2 and Table 3). As expected, there were correlations between many of the test results (Spearman’s Correlation: median 0.09; range 0.00–0.94), including between: (i) creatinine and urea, and of both with CK (range: 0.35–0.40); (ii) CK and aspartate aminotransferase (AST) (0.40); (iii) albumin and free T4, total protein and globulin (0.35–0.62); (iv) C-Reactive Protein (CRP) and Erythrocyte Sedimentation Rate (ESR) (0.47); (v) monocytes with white blood cells, lymphocytes and neutrophils (0.42–0.91) (Appendix A). In Appendix C (Table A1) we present the reference ranges for the laboratory tests, analysed in this paper. The distributions of the analytes were compared between cases and controls (Table 2), and between severe and non-severe ME/CFS cases (Table 3). ESR, platelets, and CRP were raised in PWME, while CK and urea were reduced (Table 2; Wilcoxon rank sum *p* < 0.01). Compared with mild cases, those with severe disease had raised albumin, free T4 and serum folate, and lower CK, CRP, potassium, creatinine, and bilirubin (Table 3; Wilcoxon rank sum *p* < 0.01).

The reduced CK values were mainly due to low values of CK-MM isoenzyme, for which the mean was 44 U/L (interquartile range (IQR) = 24–86) in severe cases, compared to 73 (IQR = 37–94) in the mild/moderately affected and 90 (IQR = 41–238) in healthy controls (*p* = 0.03). CK-MB and aldolase values were also lowest in severely affected cases (Kruskal–Wallis *p*-values of 0.27 and 0.09, respectively). The ratio of CK-MB/CK-total was highest in the severe 0.26 (IQR = 0.18–0.39), compared to 0.20 (IQR = 0.10–0.32) in non-severe cases and 0.14 (IQR = 0.09–0.25) in healthy controls (*p* = 0.03), confirming the main contribution of reduced CK-MM values to the low concentration of total CK in cases, and particularly the severely affected (see also Table 2 and Table 3). Total CK correlated very strongly with CK-MM (*r* = 0.99) in all study groups, but weaker correlations were found between total (CK) and CK-MB (from *r* = 0.22 in the severely affected, *r* = 0.32 in milder cases, and *r* = 0.63 in controls). This further demonstrates CK-MM as the main driver of reduced total CK concentrations in ME/CFS.

Table 4 shows results of the multivariate logistic regression, from which (ln) CK levels were shown to be inversely associated with the risk of being a ME/CFS case compared with being a healthy control (odds ratio (OR) 0.36; *p* < 0.001). The same analysis focusing on severe cases (vs. controls) revealed elevated risks associated with lower (ln) bilirubin (OR 0.23; *p* < 0.001) and lower (ln) CK (OR 0.05; *p* < 0.001), as well as with higher albumin (OR 1.20; *p* < 0.001) and T4 (OR 1.42; *p* < 0.001). Similarly, lower levels of creatinine (OR 0.91; *p* < 0.001), (ln) bilirubin (OR 0.15; *p* < 0.001) and (ln) CK (OR 0.16; *p* < 0.001) were associated with increased risk of severe compared to mild ME/CFS. Consistent with the above, increased albumin levels were associated with greater risk of being a severe compared to a mild case (OR 1.25; *p* < 0.001).

Across the analyses, differences in CK were the most consistent; there were marked differences between cases and healthy controls, with severe cases having much lower levels than mild/moderate cases or healthy controls (Figure 1a). By refitting the different outcome models either with urea, creatinine, bilirubin, albumin, T4 and CK, or with CK alone, we could estimate the value of CK as a potential marker for ME/CFS (Figure 1b). The ROC curve analysis for routine serum CK had a limited role in distinguishing ME/CFS cases from healthy controls (AUC = 67%) but had a potential role in distinguishing severely affected participants with ME/CFS from those with mild/moderate ME/CFS (AUC = 75%) and from healthy controls (AUC = 84%) (Figure 1b). A full model adding CK, bilirubin, albumin, T4, creatinine, and CRP, improved predictions, particularly when severe cases were compared to non-severe cases and healthy controls (AUC = 91%; improvement over CK alone *p* << **0.001**), and a reduced model of CK, bilirubin, albumin and T4 (based on Table 4) had similar performance (AUC = 90%).

We found a cut-off value of 51 U/L for serum CK to have high sensitivity for both ME/CFS groups of severity compared to controls (sensitivity = 96%), and for severe ME/CFS cases compared to mild/moderate cases (sensitivity = 88.7%). However, these have low specificity (<50%).

Considering the strong association between activity levels and participant category—either case or control (*p* < 0.0001)—we explored further the association between ln CK values and participant category in three different strata of activity levels, i.e., “not at all active or rather inactive”, “neither active nor inactive”, and “rather active or very active”. The results presented in Table 5, show clear trends towards lower CK values in cases, particularly as those severely-affected are compared with healthy controls. The highest significant levels are seen in those who declared they have been inactive (*p* < 0.001).

## 4. Discussion

Routine laboratory tests are usually reported as normal in PWME, when they are used to exclude other illnesses causing chronic fatigue. Among the routine laboratory tests for ME/CFS is serum CK, with studies typically reporting CK results within normal ranges if, indeed, they are reported [11,12]. We found that PWME had lower CK levels than healthy controls, and that CK was significantly lower in people with severe ME/CFS compared to those who were mild/moderately affected. The associations persisted after adjusting for sex, age-group, BMI, muscle mass, or recent physical activity levels (all of which potentially affect CK concentrations in serum [13,14]), as well as disease duration where appropriate (i.e., when comparing cases of differing severity levels). Creatinine and bilirubin were also found to be significantly lower in severely-affected patients compared to mild/moderate cases, while albumin was higher in severe cases. There was a trend towards higher levels of CRP and ESR in ME/CFS, and particularly in those with mild/moderate disease. CK levels were found to be good predictors of severe ME/CFS cases as compared to those who were less severely-affected or healthy controls; this was enhanced by the inclusion of other laboratory tests results in the model.

### 4.1. The Physiological and Clinical Importance of CK

CK is a key enzyme involved in energy production and homeostasis processes, particularly in tissues with highly dynamic and fluctuating energy demands which must be quickly satisfied, such as the brain, skeletal muscle, and heart [15]. It is also present in many other cells, such as immune and epithelial cells, where it also plays a crucial role in energy production [16,17].

Cellular energy is mainly generated from the breakdown of the high-energy molecule, adenosine triphosphate (ATP). Although the complex mechanisms of cellular energy balance are not fully understood, there is evidence that muscle cells depend on several processes to increase energy availability and to avoid complete energy depletion by drawing on stores of ATP. CK plays a crucial role in these processes: firstly, by establishing an efficient cytosolic storage of high-energy phosphates for rapid ATP replenishment [13], it catalyses the relocation of γ-phosphate from ATP to creatine to generate phosphocreatine and adenosine diphosphate (ADP), in a reversible process [18]; and secondly, it is involved in the transfer of high-energy phosphate from the mitochondria to the muscle cell cytoplasm, where it is used during muscle contraction [19]. Thus, measures of CK in serum may indicate the availability of cellular energy [13].

CK serum concentrations are widely used to diagnose and monitor a range of muscle diseases when high levels are clinically relevant. CK levels are known to increase with muscle injury and inflammation [13], and may also be high in hypothyroid myopathy and secondary to treatment with statins [14]; increased levels of serum CK can be seen in trained athletes and in individuals after extreme exercise [20]. There are three types of CK, called isoenzymes: CK-MM (found in muscle and raised in many muscle diseases), CK-MB (present in heart muscle in particular), and CK-BB (found mostly in the brain but usually undetectable in peripheral blood) [13]. CK-MB used to be widely used in the diagnosis of ischaemic heart disease, though it has now been superseded by other disease markers [21].

Low serum CK concentrations have been reported less frequently, but they may be of clinical significance in some rheumatological [22,23] and connective tissue diseases [24]. The causes and significance of these findings are unclear, but it has been suggested that serum CK may be inversely related to inflammatory processes [24,25], and low CK levels have been associated with muscle weakness, independently of muscle atrophy [23]. A study on patients with severe renal failure reported a significant association between lower serum CK and an increased risk of death, and suggested that their serological findings might be explained by the patients’ wasted muscle mass and poor nutritional status [26], but there was no subsequent suggestion of using serum CK for diagnosis or management of these conditions. In studies on Huntington’s disease, a severe degenerative neurological disease [27], it has been suggested that reduced brain energy may result from the reduced activity of the CK system, and that loss of CK-BB may be an important unrecognized biomarker of this disease.

The low concentrationsof serum CK found in people with ME/CFS suggests an abnormality in energy metabolism, which have been reported by distinct authors (e.g., references [28,29,30,31,32]) and could explain the intolerance to exertion commonly reported by patients, and consequent reduction in activity levels [2]. An alternative or additional explanation is that the lower serum CK resulted, at least partially, from physical inactivity. Nevertheless, the persistent significant association between lower serum CK and disease severity in the multivariate model that controls for activity suggests that these results cannot be fully explained by reduced physical activity, but that there are other factors involved. To explore further a potential confounding role of activity levels, we compared median serum values of CK in different strata of activity category. The values were reduced in severely affected cases in all strata Nevertheless, the potential for residual confounding is still present, and differential misclassification on activity levels, with over-estimation of activity in severe ME/CFS cases, remains a possible (and at least partial) explanation, for the difference in the levels of CK observed. However, with the increasing number of reports on energy metabolism abnormalities in people with ME/CFS, further exploration of this association is warranted.

In addition to intolerance to physical exercise, patients with ME/CFS usually also report mental fatigue and “brain fog”, as well as subjective muscle weakness [2,4,33]. Considering the importance of CK in both muscle and brain metabolism, the low concentrations of this enzyme could, at least partially, explain those symptoms.

Some researchers have previously reported intriguing findings on serum CK in PWME. For example, two studies with 30 and 33 individuals, compared the CK serum concentration of ME/CFS patients (diagnosed by the CDC-1994 criteria [3]) with those of healthy controls before and after exercise, in order to evaluate physical capacity [34,35]. The mean CK in PWME was lower, though not significantly, than in controls, and did not increase with exertion in those with ME/CFS (as seen in healthy individuals). The results suggests that lack of acute physical effort was not the main factor determining CK levels in PWME [34,35]. Another study on PWME found higher serum CK in participants with enterovirus-specific RNA detected in muscle by biopsy, than in those with no evidence of enteroviral infections. That study suggested that a sub-group of PWME might have muscle damage secondary to enterovirus infection, but unfortunately, the authors did not specify the concentrations of CK in the group of patients with lower values [36].

Other studies have considered possible mitochondrial function impairment in ME/CFS, which reinforces the plausibility of metabolic dysfunction in the energy system (e.g., references [28,29,30]) as our findings indicate.

### 4.2. Other Study Findings

Creatinine levels were found to be significantly lower in severe ME/CFS. Creatine phosphate (CP) is converted to creatinine in muscle and creatinine is excreted in urine [13], so low CP resulting from poor conversion of creatine to CP by CK could explain the low levels of creatinine found in severe ME/CFS. Low urine concentration of creatinine have also been reported [37].

We also found that urea was reduced and platelets, CRP and ESR increased in PWME, although this was not significant in the multivariate analysis. An inverse relationship between platelets and CK was found in one study on rheumatoid arthritis, together with increases in inflammatory clinical markers such as ESR and CRP [25]. Platelets also tend to increase in inflammatory processes [38]. Nevertheless, the pattern of inflammatory markers in our study leads us to speculate that some degree of inflammation may be related to symptoms in people with mild/moderate ME/CFS. For people with severe disease, it seems that the inflammatory response may have been inhibited, either due to a lack of a persistent stimuli or to impairment of some aspects of the inflammatory response, which may indicate chronic disease with established complications. However, it is important to note that changes in inflammatory markers were relatively modest, and any further assumptions would need to be appropriately tested and confirmed in a further set of samples. We found lower levels of urea in mild/moderate cases, but not in those who were severely-affected. This could relate to creatinine levels, which were also lower in ME/CFS compared to controls; however, for creatinine, the lowest values were present in the most severely-affected. Abnormalities in the urea cycle have been reported in ME/CFS (e.g., [39,40])

Other findings including raised albumin and T4 are more difficult to explain. Raised albumin suggests malnutrition is not a main factor in ME/CFS in this study. The mildly raised T4 could suggest minor changes related to thyroid hormone metabolism, such as in its peripheral conversion, as previously suggested [41]. However, this hypothesis was not corroborated by other findings, which showed similar T3 and TSH levels between the groups, neither do our findings are typical of euthyroid sick syndrome. A higher prevalence of Gilbert’s syndrome in ME/CFS has been proposed in the past [42], but this was not found in this study, which showed similar bilirubin levels in cases and controls, with reduced bilirubin levels in people with severe ME/CFS compared to non-severe cases.

Further investigations with a larger sample size and more detailed explorations of metabolic pathways will be needed to confirm whether low CK activity is a primary or secondary event in ME/CFS or, indeed, whether it reflects some other metabolic dysfunction. This could benefit from the use of “diseased” controls groups, such as selected orthopaedic patients with prolonged immobility. Meanwhile, we suggest that CK could be used as a potential marker of severe ME/CFS. It is important to consider the clinical history and physical examination findings, as well as measures of activity (e.g., outputs from accelerometers) at the time of the blood draws for serum CK. Moreover, a “high-normal” result in people who are often sedentary or bedbound, particularly those with severe ME/CFS, should be interpreted with caution, as it could indicate the presence of muscle injury.

### 4.3. Study Strengths and Limitations

The study investigated a group of people with well-characterised ME/CFS using robust standards of data and sample collection as described in the UK ME/CFS Biobank protocol [6]. In terms of ME/CFS research, this was a large sample: 272 PWME were tested, including 56 who were severely-affected. However, the assessment of activity level was by self-report [6]. If this resulted in overestimation of activity levels in some groups, then differential misclassification may have resulted e.g., if the less active and the severely-affected ME/CFS cases may have overestimated their activity levels [43].

The inclusion of severely-affected ME/CFS participants, typically absent from most previous studies, may have been key to demonstrating abnormalities not previously reported. The results presented here come from routine laboratory tests in PWME and healthy controls. The purpose of the blood tests was primarily to exclude other diseases which could present with similar symptoms; the examination of muscle-related biochemical abnormalities was not in response to a specific hypothesis but was noted when all results were examined to test null hypotheses of no group differences between study groups in laboratory test findings. This means that we were not able to examine abnormalities in muscle/energy metabolism in more detail. However, our ability to investigate our findings further, e.g., through enhanced metabolomic studies and by accessing additional biobanked samples from the cohort, including at different time-points, will be instrumental in further understanding the changes reported here.

## 5. Conclusions

This is the first study to find significant lower concentrations of CK in PWME. Some indications of low CK values seem to have been overlooked previously when the trend towards lower values was discarded as not significant and was not investigated further. A single measurement of serum CK may not have enough sensitivity and specificity to be used as a biomarker for ME/CFS diagnosis, but, used alongside other clinical and laboratory markers, routine CK blood tests could not only help to diagnose ME/CFS accurately, but also to sub-group cases according to disease severity. It could potentially also be used as a prognostic marker, and as an outcome measurement for observational studies and clinical trials, as well as in clinical practice, pending further longitudinal studies examining the correlation of clinical and laboratory-based phenotypes over time. Whether people with severe ME/CFS featuring low CK constitute a unique sub-group of patients with distinct patterns of biochemical/pathophysiological abnormalities and symptoms, or whether they represent a different phase or an extreme spectrum of the disease, still needs to be clarified.

Our findings give significant support to the growing body of evidence on metabolic abnormalities in ME/CFS, and we suggest further adequately-powered studies that include a fuller investigation of specific metabolic pathways to elucidate whether CK is a primary or secondary abnormality in all or in a sub-group of ME/CFS cases. Such studies should also help to elucidate the causes of such abnormalities. Correlating serum CK concentrations with objectively measured activity levels and other energy metabolism parameters could lead to a better understanding of the pathophysiological abnormalities involved in muscle use and recovery, and their relationship to symptoms such as post-exertional malaise and fatigue.

## Figures and Tables

**Figure 1 diagnostics-09-00041-f001:**
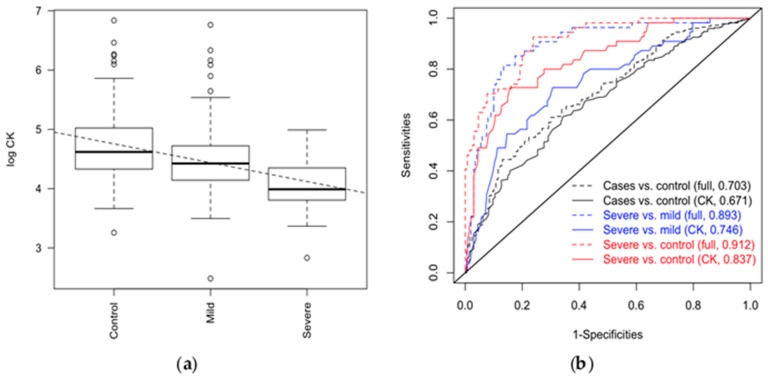
(**a**) Levels of CK and disease outcome. CK was measured in healthy controls *n* = 136, mild/moderate ME/CFS cases *n* = 216, and severe ME/CFS cases *n* = 56. Intercept 5.07411 (se 0.07907, *p* < 0.001); slope-0.31797 (se 0.04120, *p* = 0.001). (**b**) ROC curves showing the predictability of a “full” model (including urea, creatinine, bilirubin, albumin, CK, T4; dashed lined) vs CK alone (solid line); areas under the curve (AUC) are presented in the legend. CK = creatine kinase; ME/CFS = Myalgic Encephalomyelitis/Chronic Fatigue Syndrome; ROC = Receiver Operating Curves.

**Table 1 diagnostics-09-00041-t001:** Baseline characteristics of study participants.

Characteristic	Healthy Controls (*n* = 136) *n* (%)	Mild/Moderate ME/CFS Cases (*n* = 216) *n* (%)	Severe ME/CFS Cases (*n* = 56) *n* (%)	*P*-Value (hc/mm/sa)	*P*-Value (hc/me)
Female	84 (61.8)	166 (76.9)	43 (76.8)	0.006 ^a^	0.001 ^a^
Male	52 (38.2)	50 (23.1)	13 (23.2)		
Age (years)				0.900 ^a^	0.900 ^a^
18–29	23 (16.9)	34 (15.7)	10 (17.8)		
30–39	34 (25)	48 (22.3)	13 (23.2)		
40–49	35 (25.7)	64 (29.6)	12 (21.5)		
50–60	44 (32.4)	70 (32.4)	21 (37.5)		
Level of activity *					
Very active	33 (25.2)	1 (0.6)	2(3.5)	<0.0001 ^a^	<0.0001 ^a^
Rather active	59 (45)	25 (12.3)	3 (5.3)		
Neither active nor inactive	22 (16.8)	49 (24.1)	3 (5.3)		
Rather inactive	16 (12.3)	91 (44.8)	15 (26.8)		
Not at all active	1 (0.7)	37 (18.2)	33 (59.1)		
Body mass index (mean) *	25.96	27.17	23.73	0.002 ^b^	0.360 ^b^
Body muscle (%) *	49.5	46.3	45.7	0.003 ^b^	0.001 ^b^

^a^ χ^2^test; ^b^ ANOVA; *p*-values compare healthy controls with mild–moderate cases and with severe cases (penultimate column), and with all myalgic encephalomyelitis (ME) or chronic fatigue syndrome (CFS) cases (last column). * The number of participants with missing data on level of activity was 18 (5 healthy controls, 13 mild/moderate ME/CFS cases); 4 severely-affected ME/CFS cases were missing body mass index (BMI) data; and 5 severely-affected ME/CFS cases were missing data on body muscle. ME/CFS = Myalgic Encephalomyelitis/Chronic Fatigue Syndrome; hc = healthy controls, mm: ME/CFS mild/moderately affected; sa = ME/CFS severely affected; me = ME/CFS cases.

**Table 2 diagnostics-09-00041-t002:** Comparison of baseline laboratory haematological and biochemical test results from ME/CFS cases and healthy controls.

Assay	All Cases (*n* = 272) Median	IQR	Healthy Controls (*n* = 136) Median	IQR	Wilcoxon *p* ^a^
WBC (10^9^/L)	6.1	(5.2,7.3)	5.9	(5.1,6.8)	0.268
**Platelets (10^9^/L)**	**262**	**(226,310)**	**247**	**(206,282)**	**<0.001**
Haemoglobin (g/L)	137	(129,148)	139	(130,149)	0.433
Haematocrit	0.412	(0.391,0.438)	0.418	(0.395,0.446)	0.165
Neutrophils (10^9^/L)	3.47	(2.72,4.28)	3.28	(2.56,4.31)	0.374
Lymphocytes (10^9^/L)	1.88	(1.61,2.30)	1.815	(1.525,2.140)	0.146
Monocytes (10^9^/L)	0.43	(0.34,0.55)	0.43	(0.35,0.55)	0.869
Eosinophils (10^9^/L)	0.135	(0.80,0.24)	0.14	(0.09,0.22)	0.701
Basophils (10^9^/L)	0.03	(0.02,0.05)	0.03	(0.02,0.04)	0.923
**ESR (mm/h)**	**7**	**(4,12)**	**5**	**(2,8)**	**<0.001**
Sodium (mmol/L)	140	(139,142)	140	(139,142)	0.336
Potassium (mmol/L)	4.2	(4.0,4.4)	4.2	(4.0,4.4)	0.680
**Urea (mmol/L)**	**4.3**	**(3.5,5.1)**	**4.8**	**(3.9,5.7)**	**<0.001**
Creatinine (umol/L)	74	(67,85)	77	(66,88)	0.539
Adj. calcium (mmol/L)	2.36	(2.31,2.42)	2.37	(2.32,2.41)	0.900
Inorg. phosphate (mmol/L)	1.03	(0.90,1.16)	1.05	(0.93,1.14)	0.452
Total bilirubin (umol/L)	9	(7,11)	9	(7,13)	0.087
Albumin (g/L)	44	(41,47)	44	(41,47)	0.531
Globulins (g/L)	31	(29,32)	30	(28,32)	0.363
ALP (U/L)	67	(56,80)	63.5	(52,75)	0.055
AST (U/L)	20	(16,23)	20	(18,24)	0.247
Total protein(g/L)	73	(70,76)	72	(68,75)	0.023
**CK (U/L)**	**80**	**(56,107)**	**101.5**	**(76,152)**	**<0.001**
CK-MM (U/L) *	55	(28,100)	90.0	(41,238)	0.030
CK-MB (U/L) *	17	(14,22)	17	(14-23)	0.780
Aldolase *	3.8	(2.8,4.6)	4.0	(2.8,4.6)	0.620
**CRP (mg/L)**	**2**	**(1,4)**	**1**	**(1,3)**	**0.007**
Free T3 (pmol/L)	4.5	(4.1,4.9)	4.6	(4.2,5.1)	0.238
Free T4 (pmol/L)	14	(13,16)	14	(3,16)	0.242
TSH (mU/L)	1.6	(1.2,2.3)	1.62	(1.15,2.40)	0.637
Serum vitamin B12 (pg/mL)	388	(310,539)	379	(309,462)	0.164
Serum folate (ng/m)	8.6	(5.6,12.6)	9.0	(6.6,12.2)	0.313

^a^ Wilcoxon rank sum test; * For CK isoenzymes CK-MM and CK-MB and aldolase, *n* = 50 cases and 25 controls. WBC = white blood cells; ESR = erythrocyte sedimentation rate; Adj. calcium = adjusted calcium; Inorg. phosphate = inorganic phosphate; ALP = alkaline phosphatase; AST = aspartate aminotransferase; CK = creatine kinase; CK-MM = CK produced in skeletal muscle; CK-MB = CK produced in heart muscle; CRP = C-reactive protein; TSH = thyroid stimulating hormone; IQR = interquartile range. In bold are variables (*p* < 0.001) carried forward for regression analysis.

**Table 3 diagnostics-09-00041-t003:** Comparison of baseline laboratory haematological and biochemical test results from severe and mild/moderate ME/CFS cases.

Assay	Severe Cases (*n* = 56) Median	IQR	Mild Cases (*n* = 216) Median	IQR	Wilcoxon *p ^a^*
WBC (10^9^/L)	5.85	(5.05,7.54)	6.11	(5.2,7.1)	0.744
Platelets (10^9^/L)	254.5	(221,305)	266	(229,310)	0.585
Haemoglobin (g/L)	134	(126,144)	137.5	(130,149)	0.038
Haematocrit	0.405	(0.388,0.428)	0.413	(0.391,0.440)	0.156
Neutrophils (10^9^/L)	3.44	(2.89,4.20)	3.485	(2.68,4.32)	0.856
Lymphocytes (10^9^/L)	1.785	(1.59,2.05)	1.915	(1.62,2.32)	0.061
Monocytes (10^9^/L)	0.46	(0.36,0.59)	0.425	(0.34,0.54)	0.078
Eosinophils (10^9^/L)	0.165	(0.075,0.265)	0.13	(0.08,0.23)	0.366
Basophils (10^9^/L)	0.03	(0.02,0.05)	0.03	(0.02,0.04)	0.697
ESR (mm/h)	5	(2,10)	7	(5,12)	0.057
Sodium (mmol/L)	141	(140,142)	140	(139,141)	0.015
**Potassium (mmol/L)**	**4**	**(3.8,4.3)**	**4.2**	**(4.0,4.4)**	**0.003**
Urea (mmol/L)	4.15	(3.4,5.1)	4.3	(3.5,5.2)	0.317
**Creatinine (µmol/L)**	**65**	**(59,74)**	**78**	**(68,86)**	**<0.001**
Adj. calcium (mmol/L)	2.35	(2.29,2.41)	2.37	(2.31,2.42)	0.048
Inorg, phosphate (mmol/L)	1.03	(0.93,1.16)	1.03	(0.89,1.16)	0.439
**Total bilirubin (µmol/L)**	**7**	**(5,9)**	**9**	**(7,12)**	**<0.001**
**Albumin (g/L)**	**46**	**(44,49)**	**43**	**(40,46)**	**<0.001**
Globulins (g/L)	29	(29,29)	31	(29,33)	0.542
ALP (U/L)	64	(54,80)	67	(56,81)	0.388
AST (U/L)	19	(16,22)	20.5	(16.5,24.0)	0.438
Total protein(g/L)	68	(68,68)	73	(70,76)	0.148
**CK (U/L)**	**54**	**(45,78)**	**84**	**(63,113)**	**<0.001**
CK-MM (U/L) *	44	(24,86)	73	(37,194	0.030
CK-MB (U/L) *	16	(13,20)	19	(17,22)	0.080
Aldolase *	3.6	(2.5,4.0)	4.0	(2.8,4.6)	0.040
**CRP (mg/L)**	**1**	**(1,5)**	**2**	**(2,6)**	**<0.001**
Free T3 (pmol/L)	4.4	(4.1,4.7)	4.6	(4.2,4.9)	0.073
**Free T4 (pmol/L)**	**16.1**	**(14.2,18.0)**	**14.0**	**(13.0,15.3)**	**<0.001**
TSH (mU/L)	1.59	(1.14,2.28)	1.61	(1.16,2.40)	0.795
Serum vitamin B12 (pg/mL)	449	(333,659)	382	(305,532)	0.038
**Serum folate (ng/m)**	**9.8**	**(8.0,14.7)**	**7.6**	**(5.0,12.6)**	**0.002**

^a^ Wilcoxon rank sum test; * For CK isoenzymes CK-MM and CK-MB and aldolase, 25 cases in each group. WBC = white blood cells; ESR = erythrocyte sedimentation rate; Adj. calcium = adjusted calcium; Inorg, phosphate = inorganic phosphate; ALP = alkaline phosphatase; AST = aspartate aminotransferase; CK = creatine kinase; CK-MM = CK produced in skeletal muscle; CK-MB = CK produced in heart muscle; CRP = C-reactive protein, TSH = thyroid stimulating hormone, IQR = interquartile range. In bold are variables (*p* < 0.01) carried forward for regression analysis.

**Table 4 diagnostics-09-00041-t004:** Laboratory test results in ME/CFS cases and healthy controls (adjusted analysis ***).

	Assay	Odds Ratio	(95% CI)	*p*-Value
ME/CFS cases vs. healthy controls	Platelets	1.01	(1.00–1.01)	0.007
	Urea	0.78	(0.65–0.94)	0.008
	**CK ***	**0.36**	**(0.23–0.57)**	**<0.001**
	CRP *	1.37	(0.99–1.90)	0.059
	ERS **	1.33	(1.04–1.71)	0.023
Severe ME/CFS cases vs. healthy controls	Urea	0.71	(0.53–0.96)	0.027
	Creatinine	0.95	(0.91–0.98)	0.001
	**Bilirubin ***	**0.23**	**(0.10–0.52)**	**<0.001**
	**Albumin**	**1.20**	**(1.08–1.33)**	**<0.001**
	**CK ***	**0.05**	**(0.02–0.15)**	**<0.001**
	**T4**	**1.42**	**(1.21–1.66)**	**<0.001**
	Vit. B12 *	4.12	(1.64–10.36)	0.0026
Severe ME/CFS cases vs. mild/ moderate ME/CFS cases ****	Potassium	0.38	(0.13–1.13)	0.082
	**Creatinine**	**0.91**	**(0.87–0.94)**	**<0.001**
	**Bilirubin ***	**0.15**	**(0.06–0.36)**	**<0.001**
	**Albumin**	**1.25**	**(1.13–1.38)**	**<0.001**
	**CK ***	**0.16**	**(0.07–0.40)**	**<0.001**
	CRP *	0.56	(0.32–0.97)	0.040
	T4	1.26	(1.09–1.45)	0.0014
	Folate *	1.83	(0.98–3.40)	0.058

* ln transformed; ** Square-root transformed; *** Adjusted for sex, age-group, BMI, and muscle mass, current physical activity level; **** Further adjusted for disease duration; ME/CFS = Myalgic Encephalomyelitis/Chronic Fatigue Syndrome; ESR = erythrocyte sedimentation rate, CK = creatine kinase, CRP = C-reactive protein. Bolded text signifies *p* < 0.001.

**Table 5 diagnostics-09-00041-t005:** Median total CK (IQR) according to reported activity levels *.

	Inactive (*n* = 190)	Average (*n* = 72)	Active (*n* = 121)
Healthy controls (*n* = 129)	76(70,157)	98(66,152)	110(84,153)
Mild/moderate ME/CFS cases (*n* = 199)	82(61,110)	92 (68,143)	85(73,108)
Severe ME/CFS cases (*n* = 55)	52(45,77)	46 (44,78)	60(46,91)
*p*-value **	<0.001	0.200	0.020

* Association between activity levels and case category (chi-squared test) *p* < 0.0001; ** *p*-value for Kruskal–Wallis statistics comparing CK in healthy controls, mild-moderate, and severe cases within each activity level group.

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
