# Peer review of "Evidence of Clinical Pathology Abnormalities in People with Myalgic Encephalomyelitis/Chronic Fatigue Syndrome (ME/CFS) from an Analytic Cross-Sectional Study"

_diagnostics, 2019, doi:10.3390/diagnostics9020041_

Round 1
Reviewer 1 Report
Table 1A Supplementary: Should not the normal range for CK be <320U/L not >320U/L as shown in Table?
Author Response
We have corrected it. Thank you very much.
Reviewer 2 Report
Revisions are acceptable
This manuscript is a resubmission of an earlier submission. The following is a list of the peer review reports and author responses from that submission.
Round 1
Reviewer 1 Report
This manuscript describes results of 19 routinely measured clinical pathology parameters in well characterised cohorts of patients with severe ME/CFS (n=56) vs moderate ME/CFS (n=216) vs age matched healthy controls (n=136). The results and conclusions are clearly set out in the abstract. Conclusions were that after adjusting for a number of relevant variables, including in particular activity levels and BMI, CK levels in severe group were significantly lower than either those with moderate disease and healthy controls. This is a simple but extremely well controlled study of serum variables in ME/CFS patients using samples from a thoroughly characterised patient cohort, matched with healthy controls for comparison.
The body of the manuscript however is difficult to follow and needs editing. In particular, the data presentation is overly complicated and gives no real sense of the distribution of results in relation to the normal ranges encountered within the clinic. Non-parametric statistics are usually used for this sort of data throughout so the reader can relate to what is usual in research and clinical practice and also for consistency within the manuscript. Transformation may of course be necessary for multivariate analysis to take into account confounding factors to determine sensitivity of CK comparing severe ME/CFS patients with other groups.
Some simple plots of the raw data for variables of interest (eg CK) in relation to normal ranges would provide a more solid platform for interpretation of statistical methods subsequently applied.
The methods and results contain extensive detail – more than subsequently justified by data shown.
Although of potential for mechanistic research based on these preliminary data, with respect to being a possible biomarker: It is not clear whether despite the median CK in severe patients being lower than that of moderate patients, results still seem to all be within the normal range. How is this to be used in practice or as a basis for stratification? For example: could the authors supply a number and range, based on their calculations, below which severe disease could be confirmed?
Specific points:
Table 1: Statistical test not given (? Presumably ANOVA) – would also be more useful comparing all cases with HC to see the goodness of ‘matching’ between patient group as a whole and healthy controls. Also layout: P values - not matching lines of data so not able to determine what they are referring to clearly enough. Very little in text.
Tables 2 and 3: Test cited is Wilcoxon. Surely this is only applicable to paired results which this data is not? Thus not sure what the p values actually mean. In house ranges for variables should be given.
Table 4: Adjusted data sufficient as univariate results already dealt with.
Figure 1: log CK. Had to assume ‘natural log’ was the scale as not clear. Log usually means Log10 so ln should be used. Mentioned in text but only once and subsequent transformed data not defined as natural logs either. Same for Table 5.
Table 5; Ln numbers are also very hard to make sense of when dealing with numerical values which are used in both routine or research scenarios. P values only compare cases. Comparisons of cases with HC mentioned in text but not given.
· Isoenzymes are mentioned in Discussion but no results included. This would strengthen data set.
· Discussion is overly long and speculative includes a wide-ranging discussion of the condition outside the scope of the results, being more suited to an (interesting) review.
Author Response
Comment: The body of the manuscript however is difficult to follow and needs editing. In particular, the data presentation is overly complicated and gives no real sense of the distribution of results in relation to the normal ranges encountered within the clinic. Some simple plots of the raw data for variables of interest (e.g. CK) in relation to normal ranges would provide a more solid platform for interpretation of statistical methods subsequently applied.
Response: We thank you for your comments, which have helped us to further clarify the reporting of our findings. We have now extensively edited the paper to add clarity and avoid redundancy. As suggested, we present plots with CK results as a supplementary figure (Appendix A), and normal ranges for the laboratory tests analysed in the paper as an Appendix table (Appendix B – Table 1a).
Comment: Although of potential for mechanistic research based on these preliminary data, with respect to being a possible biomarker: It is not clear whether despite the median CK in severe patients being lower than that of moderate patients, results still seem to all be within the normal range. How is this to be used in practice or as a basis for stratification? For example: could the authors supply a number and range, based on their calculations, below which severe disease could be confirmed?
Response: The CK distribution shows that patients with more severe symptoms tend to have lower levels of CK compared to the distribution of those with mild/moderate symptoms and healthy controls. There is variation in what is considered a low concentration in CK (NB we only provide abnormal results when higher than 320 U/L). We propose a cut-off point of 51 U/L, with values below that having a 96% sensitivity to identify severe cases. This is presented in lines 198-201. However, while sensitivity is high, specificity is low, which means that although values below 51 will provide a group of people including people with severe ME/CFS, further assessment of patients is required to confirm severe disease or otherwise. Such assessment should include the assessment of disease severity indicators from clinical history and physical examination together with information on activity levels. We consider that the combination of CK serum concentration and clinical assessment tools will greatly help clinicians (and researchers) to stratify individuals with ME/CFS according to severity.
Specific points:
Comment: Table 1: Statistical test not given (? Presumably ANOVA) – would also be more useful comparing all cases with HC to see the goodness of ‘matching’ between patient group as a whole and healthy controls. Also layout: P values - not matching lines of data so not able to determine what they are referring to clearly enough. Very little in text.
Response: Thank you for pointing that out. We have provided the corresponding test carried out for each line in the Table 1 footnote and aligned the P value that was misplaced, taking the opportunity to correct some typos in Table 1. We also added normal reference population values in Appendix B, as well as the P-values when comparing all cases with healthy controls. The total number of cases is easily obtained by adding mild/moderate with severe cases. The test comparing continuous variables is indeed the ANOVA.
Comment: Tables 2 and 3: Test cited is Wilcoxon. Surely this is only applicable to paired results, which this data is not? Thus not sure what the p values actually mean. In house ranges for variables should be given.
Response: Thank you again for spotting that potential confusion. We confirm that the data are not paired and this is why we have used Wilcoxon ranksum test, which is the recommended test for the unmatched data (Kirkwood B, Sterne J. Medical Statistics. Second ed.: Blackwell; 2005). This is now fully referenced in the text (lines 96, 137, 138, and 139, and Tables 2 and 3).
Comment Table 4: Adjusted data sufficient as univariate results already dealt with.
Response: We now present adjusted data only.
Comment Figure 1: log CK. Had to assume ‘natural log’ was the scale as not clear. Log usually means Log10 so ln should be used. Mentioned in text but only once and subsequent transformed data not defined as natural logs either. Same for Table 5.
Response: That is correct, we use the natural log. This has been changed throughout the text and tables. Thank you for helping us to avoid any confusion.
Comment Table 5; Ln numbers are also very hard to make sense of when dealing with numerical values, which are used in both routine or research scenarios. P values only compare cases. Comparisons of cases with HC mentioned in text but not given.
Response: We re-structured the table to present CK medians, instead of Ln (CK). The P-value results from the analysis according to levels of activity; they do indeed represent the comparison of the 3 groups (healthy controls, mild/moderate, and severe cases) compared using Kruskal-Wallis test. We mention this in the text. We adjusted the text, by breaking the relevant sentence (lines 205 to 208) into two, to make this clear. Now it reads: “The results presented in Table 5 show clear trends towards lower values in cases, in particular those severely affected, when compared to controls. The highest significant levels were in those who declared they were inactive (P<0.001)”.< p="">
Comment: Isoenzymes are mentioned in Discussion but no results included. This would strengthen data set.
Response: We have now included the results for isoenzymes for a sub-sample of 50 cases (25 in each of the severity categories) and 25 controls. This is shown in lines 73 to 74; 140 to 150 and Tables 2 and 3.
Comment: Discussion is overly long and speculative includes a wide-ranging discussion of the condition outside the scope of the results, being more suited to an (interesting) review.
Response: We have now substantially simplified the discussion.
Reviewer 2 Report
This manuscript contains no biologically plausible hypothesis for CFS/ME. The authors provide little rationale for their approach and claims in this current manuscript.
The statement 'Although dysregulation of the nervous, immune and 40 endocrine systems, with impaired cellular energy metabolism and ion transport, has been suggested 41 [4], the pathophysiology of ME/CFS is still not well understood, and there are no biological markers 42 that can be used for diagnosis, disease sub-grouping or prognosis.' fails to grasp emerging science in this area.
The authors do not cite adequately current literature in this area, rather only provide one reference. This reflects their limited understanding of the current research.
Creatine kinase was found lower in people with severe CFS/ME than mild or health controls. Given CK is associated with inactivity was a control group considered of other clinical conditions associated with prolonged inactivity? e.g. long duration orthopaedic cases not likely to be confounding clinically. There was no group to be included that should have controlled for this confounder. This claim for CK as biomarker is unsupported.
Measuring CK before and after exertion in CFS/ME patients is dubious ethically and experiments requiring forced exercise are widely condemned by patients and do not add to any hypothesis.
The authors do not provided data for NK activity- a key feature of the ICC as well as for CFS/ME patients to be classified as CFS/ME.
Author Response
Comment: The authors do not cite adequately current literature in this area, rather only provide one reference.
Response: We have added references to the relevant literature, including energy metabolism changes in ME/CFS, e.g. references 28 to 32.
Comment: The paper “contains no biologically plausible hypothesis for CFS/ME”:
Response: Our null hypothesis was of no group differences between study groups in laboratory test findings, and this is extensively discussed throughout the text (please also see below). However, the significant contribution of this paper is that, contrary to the general belief that laboratory tests are usually normal in people with ME/CFS, we found some abnormalities, which could be very useful in both clinical practice and research, e.g. for case stratification. We are very clear on the nature and objectives of the paper, and we specifically mention this under “study strengths and weaknesses”. Lines 342 to 346 read: “The purpose of the blood tests was primarily to exclude other diseases which could present with similar symptoms; the examination of muscle-related biochemical abnormalities was not in response to a specific hypothesis, but was noted when all results were examined to test null hypotheses of no group differences between study groups in laboratory test findings”.
As is usually the case in research, evidence grows from exploratory studies to hypothesis driven studies. This study, with all importance that it has on its own for clinical practice and research, is key to generating new hypotheses. We offer a clear plan to test this new hypothesis. Lines 354-357, now modified for further clarity read: “However, our ability to investigate our findings further, e.g. through enhanced metabolomic studies and by accessing additional biobanked samples from the cohort, some at different time-points, will be instrumental in further understanding the changes reported here.”
Comment: The statement 'Although dysregulation of the nervous, immune and 40 endocrine systems, with impaired cellular energy metabolism and ion transport, has been suggested 41 [4], the pathophysiology of ME/CFS is still not well understood, and there are no biological markers 42 that can be used for diagnosis, disease sub-grouping or prognosis.' fails to grasp emerging science in this area.
Response: Despite the recent increase in research funding in this area, and the attendant increase in good quality papers enabling a somewhat better understanding of pathophysiological mechanisms in ME/CFS, we reason that there is still important evidence missing. The mounting evidence may eventually provide clear diagnostic tools and lead to effective therapeutic options, but these are not available yet. In the interest of scientific impartiality in line with international practice, we do not consider there is adequate biomarker evidence for immediate clinical and research use in ME/CFS, although results of tests such as those presented here and others (e.g. immunological tests) can be used to support diagnosis if used alongside robust clinical assessment.
Comment: Given CK activity is associated with inactivity was a control group considered of other clinical conditions associated with prolonged inactivity? e.g. long duration orthopaedic cases not likely to be confounding clinically. There was no group to be included that should have controlled for this confounder. This claim for CK as biomarker is unsupported.
Response: Indeed, we agree that orthopaedic controls might be useful, in relation to inactivity, but we did not recruit such patients. This could be a very useful comparison group in future studies to confirm or negate our results (we suggest this for further studies lines 347 to 350). We are not claiming that CK abnormality explains ME/CFS or that CK should be used as a biomarker for ME/CFS by itself. Accordingly, our conclusion is much more measured. We suggest that routine CK levels in combination with other clinical and laboratory markers could help in the diagnosis of ME/CFS, and may also help to sub-group cases according to disease severity (please see lines 354 to 357).
Comment: Measuring CK before and after exertion in CFS/ME patients is dubious ethically and experiments requiring forced exercise are widely condemned by patients and do not add to any hypothesis
Response: We have not done this as part of our study. However, exercise studies have been widely used in ME/CFS research, and there is a growing number of pre- and -post exertion studies contributing to the evidence; we believe that these can be undertaken ethically without detriment to the participants, as long as full ethical procedures are scrupulously followed and consent is fully informed. However, we decided to remove this suggestion to avoid any controversy.
Comment: The authors do not provided data for NK activity- a key feature of the ICC as well as for CFS/ME patients to be classified as CFS/ME.
Response: The ICC propose clinical diagnostic criteria and their report references studies showing decreased NK activity, alongside other potential abnormal immune responses to infections, to conceptualise the clinical symptoms. In our view the “key features” in that document include the proposed group of symptoms that may indicate a diagnosis of ME/CFS. Furthermore, there is no recommendation as part of the ICC to run NK cell tests routinely for ME/CFS diagnosis, or as part of currently accepted clinical diagnostic criteria for ME/CFS. NK activity is not part of our routine laboratory analysis, but we are carrying out a study with NK cells, which is part of a distinct publication (Cliff J et al, Frontiers in Immunology, in press).